# Green Roof Enhancement on Buildings of the University of Applied Sciences in Neubrandenburg (Germany) in Times of Climate Change

**Manfred Köhler** [1],[*] and **Daniel Kaiser** [2]

1 University of Applied Sciences Neubrandenburg, Brodaer Str. 2-4, 17033 Neubrandenburg, Germany
2 Lehr- und Versuchsanstalt für Gartenbau und Arboristik e.V., 14979 Großbeeren, Germany; kaiser@lvga-bb.de
* Correspondence: Koehler@hs-nb.de

**Abstract:** The reduction in evaporative surfaces in cities is one driver for longer and hotter summers. Greening building surfaces can help to mitigate the loss of vegetated cover. Typical extensive green roof structures, such as sedum-based solutions, survive in dry periods, but how can green roofs be made to be more effective for the longer hot and dry periods to come? The research findings are based on continuous vegetation analytics of typical extensive green roofs over the past 20 years. -Survival of longer dry periods by fully adapted plants species with a focus on the fittest and best adapted species. -Additional technical and treatment solutions to support greater water storage in the media in dry periods and to support greater plant biomass/high biodiversity on the roofs by optimizing growing media with fertilizer to achieve higher evapotranspiration (short: ET) values. The main findings of this research: -The climate benefits of green roofs are associated with the quantity of phytomass. Selecting the right growing media is critical. -Typical extensive green roof substrates have poor nutrition levels. Fertilizer can significantly boost the ecological effects on $CO_2$ fixation. -If the goal of the green roof is a highly biodiverse green roof, micro-structures are the right solution.

**Keywords:** extensive green roofs; climate change; summer drought; urban vegetation; phytomass; fertilizer; biodiversity; blue green infrastructure

## 1. Introduction

Drought is related to massive deforestation in many parts of the world [1]. The daily new ground sealing around the world has not stopped, but new tree plantation initiatives work against this trend to stop further drying out of our planet. Successful initiatives of plantings are active in Australia, under the term of "land-care" [2]; Mongolia [3]; and the current Green Wall movement [4] in North Africa. There is still a long way to go until forest cover values, such as those that existed in central Europe 2000 years ago, of above 80% are reached, in contrast to today with about 30% forest cover [5]. The increasing world population is one reason for deforestation [6]. Drought in cities is connected with the low amount of evaporative green areas in cities [7,8]. The consequences are longer hot and dry summer periods that cause a number of environmental and health problems for residents. According to Kravcik et al. [9], evaporative surfaces are key instruments to combat global warming in cities. Wherever it is possible, urban forestry should be the first choice to enhance ecosystem services by planting trees, like the "trillion tree initiative" [10]. However, the second best choice is to green building surfaces, which are normally un-vegetated.

Green roofs and vegetated green facades are tools to support decentralized local water cycles and are widely used to combat the urban heat island effect. The current situation about green roofs shows that only a small amount of roofs are greened. Most of them are shallow growing media with about a 10-cm depth. Today, there is a gap between the potential greening of roofs and the number of projects that have currently been

realized. Currently, the cities with the highest rates of roof top greenery are Singapore [11], Chicago [12], and the German cities Stuttgart and Berlin [13]. The coverage rates of green roofs are between 3% and 8%, while vertical green systems, also known as living walls or green facades, are well below 1% green coverage of all buildings [14]. However, the potential for roof greening is up to 50% of all buildings. Today, in the search for more solutions in cities to deal with urban drought and provide more cooling strategies to mitigate the increase in extreme high temperature values, green roofs can potentially function as spaces to cool down urban surfaces.

Green roofs are instruments with many additional benefits, which are supported by the structures of growing media, an extra drainage layer, and vegetation cover [15]. The extra urban green space that becomes available for recreation and the greater biodiversity are further positive reasons to invest in such technologies. Cristiano et al. [16] conducted a literature review of the water–energy–food–ecosystem nexus, pointing out the multiple benefits of green roofs in contributing to the sustainable development goals of the United Nations.

The current situation in Germany is 15% coverage with intensive roof gardens and about 85% coverage with extensive green roofs [13]. Extensive green roofs normally have low rates of evapotranspiration in summer due to the low water storage capacity in the typical 10-cm-thick layer of growing media. This report highlights some details of the construction (growing media and retention layer) as well some treatments to optimize the functionality of extensive green roofs in the future [17].

The thesis in this publication is the need to shift from extensive green roofs that can survive long dry periods but that only have low evapotranspiration rates to semi-intensive green roofs with better ecological performance that form part of the blue-green infrastructure in cities [18]. Green roofs are underestimated in their potential to contribute against climate change. Cock and Larson summarized 179 peer-reviewed surveys to provide evidence of the multi-disciplinary ecological functionality. They concluded that greater efficiency is possible with some updates in some roof construction details [19].

In a meta-study, Shafique et al. [20] analyzed the efficiency of $CO_2$- sequestration of green roofs. They found two effects: a direct effect, such as uptake via photosynthesis, and an indirect effect, like extra insulation value for the building. Both have lower heating/cooling demands as a positive consequence. Most of these surveys were based on model calculation, and more local experimental works over longer periods are recommended. The thesis in this report is that more evaporation is connected with more phytomass in general, and this can have positive effects on greater $CO_2$ fixation; what kind of effect will have this on the other aim of greater plant biodiversity?

The tests in this publication involved varying the types of growing media and media depth:

-Is more evaporative phytomass produced on green roofs by selecting the right growing media and adapted plant species?

-Does irrigation or fertilization have positive effects? How do these treatments influence the goal of greater plant biodiversity on roofs? There are two theses: on the one hand, more fertilizer support more phytomass, but what about the rare species: do they prefer poor media? Is a lot of biomass counterproductive to biodiversity?

-How much better does thicker growing media for green roofs perform concerning $CO_2$ fixation as an indicator [21]? Today, solutions for more opportunities for $CO_2$ fixation are searched for.

## 2. Experiments

Two buildings of the University of Applied Sciences in Neubrandenburg were constructed as research and demonstration roofs. Building 2 of this campus complex was opened in 1999 with a research green roof site of about 2000 m$^2$ while building 3 opened in 2001 with around 1000 m$^2$ of green roof space. The details of the roofs can be seen on

Google maps with the GPS Coordinates: Degree:Minutes:Seconds); see the sites: building 2: 53:33:23N 13:14:44E and building 3: 53: 33:15N 13:14:43E.

Preliminary reports presented the results of the continuous 20-year climate measurements on these roofs. The evidence of the positive influence of the microclimate of the vegetation layer is seen in a significant reduction in the surface temperature [22]. A further paper explains how different retention layers can capture larger volumes of rain in a comparison between conventional growing media and various retention layers. This also contributes to greater and more sustained evapotranspiration to mitigate summer heat and reduces run-off from the building. In the optimized cases, the run-off from a building with a green roof can approach zero [23].

This paper presents the results for the increased efficiency of green roofs as a result of the amount of phytomass produced as an indicator of greater storage of $CO_2$, a measurable variable in times of climate change. The plant performance on a typical 10-cm-thick layer of growing media was compared to a thicker layer of 30 cm (see Figure 1a,b).

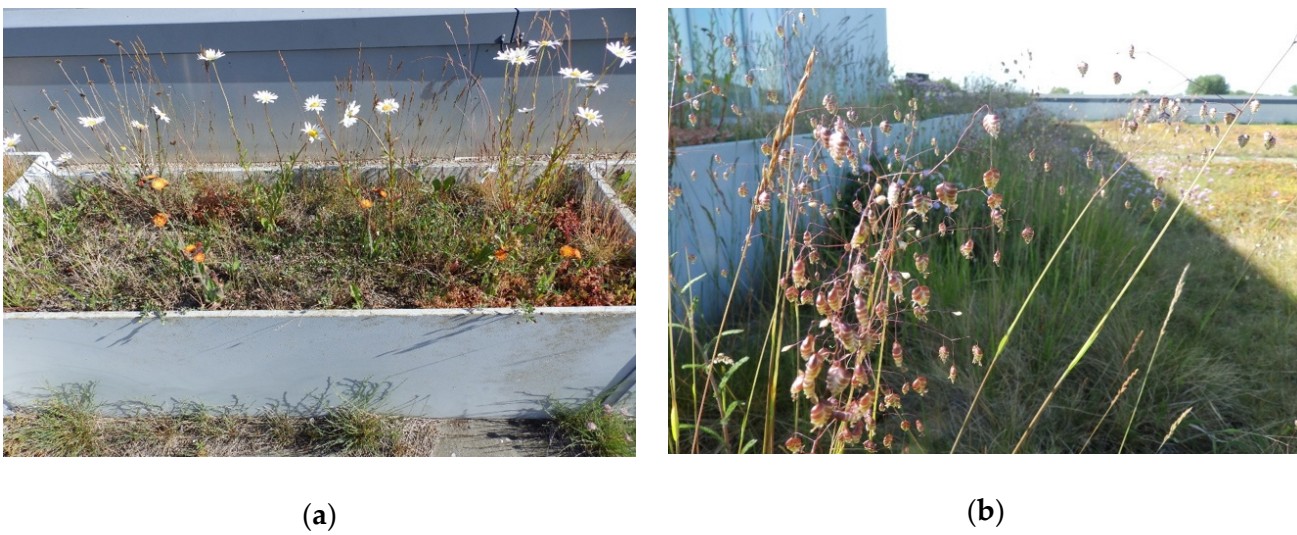

(**a**)　　　　　　　　　　　　　　　　　　　　　　　　　　(**b**)

**Figure 1.** (**a**) Example of one of the 39 planter boxes with the 30-cm growing media. (**b**) One of the microhabitat structures on the roof of building 2, the west part of the green roof. The semi-shade situation supports some endangered plants, such as the grass *Briza media*, enabling it to survive on the extensive 10-cm roof media for 20 years.

**Research design**

In contrast to many other green roof research studies, the focus here was the long-term performance of typical FLL-Standard [24] green roofs under real roof conditions. The second aspect was a complete green roof not a small test installation. Easy accessibility and the inclusion into teaching programs has allowed frequent observation of changes. The use of market leader products, such as in the German FLL guidelines, which have existed since 1990, has helped to improve these technical standards. Additionally, some further growing media and treatments, such as irrigation and fertilization, has helped to extend the existing knowledge. At the beginning, in 1999, the basic aims were to achieve 60% vegetation cover on a very shallow layer. In the last years, new upcoming questions were the enhancement of biodiversity and $CO_2$ fixation under hotter and dryer summer periods in central Europe. The basic research design can be described on both buildings as follows:

-Building 2 had three commercial growing media, identified as Zi, Blä, and Op, with a 10-cm depth plus a drainage layer. The additional 26 planter boxes with a 30-cm-thick layer of the extensive growing media from two main deliverers of green roof materials, ZinCo and Optigruen with the abbreviation Zi and Op, were the test installations. These materials followed the necessary requirements after [16], such as a granulometric distribution and the minimum water holding capacity. The primary vegetation was similar, using grass seeds and sedum cuttings.

-Building 3 had the 10-cm growing media Op-2 (product name "Optima Tiefgarage schwer") and Ulo (expanded slate, grain size 2–11 mm, brand name "Thüringer Bläh-schiefer"). The vegetation layer on top of the growing media was prefabricated turf mats on the west, north, and south sides while the east side used both growing media with a selection of sedum cuttings on top. An additional 13 planter boxes with a 30-cm media depth of Op-2 were used, representing semi-intensive green roofs with some grass seeds and some herbal plantings in the beginning.

**Research methods**

In the years up to 2010, the team investigated the floral and vegetative components of the test plots using methods derived from the Braun-Blanquet methods [25]. On each sub-plot of the roof, a complete species list was produced by noting the percentage of the area covered by each vascular higher plant genus. Additional information included the total coverage value of the indicator group of sedum and grasses.

The green roof turf mats on building 3 were a pre-grown professional turf layer with a mixture of grasses and sedum species. Such industrially produced mats have been used in many comparable projects throughout Germany. The plant species development is a role model for many typical extensive green roofs. On building 3, one question examined how these well-suited vegetation layers performed compared to green roofs started with sedum cuttings only, as an alternative approach to provide a cheaper vegetation layer on flat roofs. In the following text, the west, north, and south sides with turf mats are compared to the east side with the sedum cuttings (see Figure 2a,b).

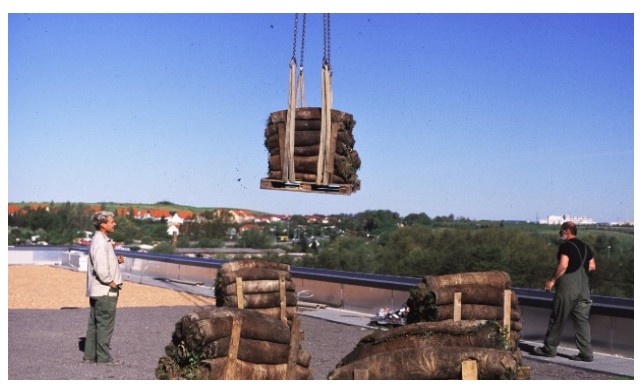
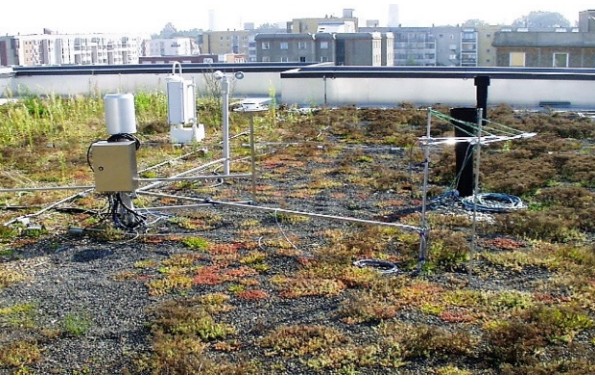

(**a**)                                                          (**b**)

**Figure 2.** (**a**) Building 2, May 2001: The turf mats were placed on the north plots, as was done on the west and south plots. The growing media can be seen in the background: light red Op, dark in the front: Ulo. (**b**) Sedum cutting area on the east roof in August 2002: Right: Ulo, sedum developed slowly, left: Sedum and several spontaneously sown grasses covered the Op successfully. Maintenance work on the weather station caused some open areas in the vegetation cover.

In contrast to this, a vegetation layer completely based on sedum cuttings and grass seeds was constructed on building 2 in 1999.

The test plots on building 3 were divided in half, with one half using the professional Op growing medium (Optima-extensive media) and the other half Ulo (Ulopor—expanded slate 2–11 mm sizes) growing medium. Op, similar to the Zi (Zinco growing media on building 2), is a professional growing medium that satisfies all the growing media requirements of FLL 2018 – Standard [24] in regard to the water retention capacity, grain size distribution, and basic fertilizer. On building 2, the Blä substrate was used as a test. This is simply crushed expanded clay as a recycled product with no additional nutrition, merely being a supplement for extensive green roofs. The same applies to Ul (Ulopor). This expanded slate has good performance in terms of its water retention capacity, but its dark color means that it heats up considerably in summer, much more than the typical

gravel layer of 16/32-mm stones. On the east side the vegetation reached the maximum coverage of 80% on Op and 75% on Ulo in 2002, the second year. This was ultimately a comparison between the different layers to learn more about the long-term performance of the materials for improving the growing media mixtures and to use the results to contribute to the updates of the FLL guideline [24].

**Harvest on 10-cm media**

In 2017, from the growing media test plot Zi, Blä, Op, Ulo, and Op-2, all above-ground and root mass was harvested, with three replicates for each test plot. All phytomass material was divided into the following groups: above-ground vascular plants shoots, roots, mosses, and lichens. The material was dried to a constant weight. The total carbon was 50% of the dry mass. The $CO_2$ concentration was calculated by multiplying with 3.65 to provide a guide value [26–29].

**Harvest on 30-cm media**

On building 2, the 26 planter boxes on the roof terraces were used as phytomass test plots. Using scissors, the above-ground phytomass was harvested in 2011, 2012, 2013, 2014, 2015, and 2017. The similar 13 boxes on building 3 were treated in the same way in 2013, 2014, 2015, and 2017. The oven-dried material allowed an initial estimate of the annual growth rate on these materials.

**Fertilizer and Irrigation**

Since 2011, on building 3, subplots were established, as shown in Table 1, to investigate the effects of additional irrigation and subplots of the same size were established to investigate the effects of fertilizer use. These subplots were compared with ongoing monitoring of the plant lists in the normal roof positions.

**Table 1.** The sizes of the 10-cm depth research subplots on building 3 from 2001; on each part of the roofs, the following sized areas were selected for treatments with additional irrigation and fertilizer, and the "normal" comparison plots for the four test plots (in $m^2$).

| Exposure | Media 1: Ulo | Media 2: Op-2 |
|---|---|---|
| West | 8 | 12 |
| North shaded | 6 | 6 |
| North sun | 6a | 6 |
| South | 5.5 | 5.5 |
| East | 11 | 11 |

The different sizes are due to the roof construction.

Additional irrigation was applied once a year in May using 10 L/$m^2$ tap water. The fertilization subplots were fertilized with 10 g/$m^2$ Kristalon 16-11-16 NPK plus MG on the same day. Figure 3a shows the effect of the fertilizer on the eastern part of the roof and Figure 3b gives an impression of the western part with the sedum mats. *Sedum sexangulare* was stimulated to flower intensively while *Allium schoenoprasum* and *Petrorhagia saxifraga* showed significantly better coverage and performance.

**Statistics**

For the statistics, SPSS_Vers.27 procedures, such as descriptive statistics, dependent *t*-tests with two variables each, tests of significance, cluster analysis, and analysis of variances (ANOVA), were completed using the datasets.

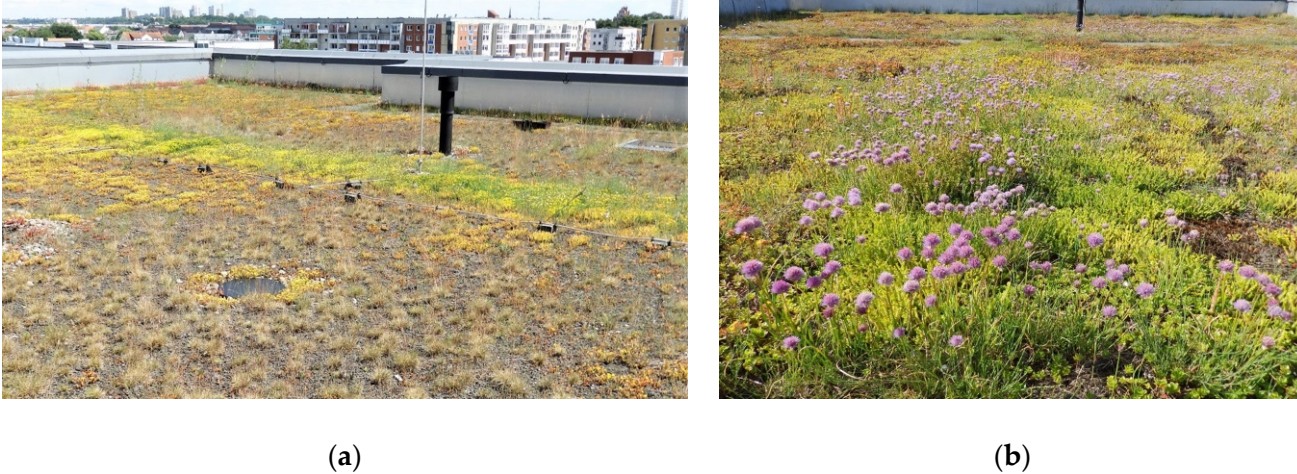

|                    (a)                    |                    (b)                    |

**Figure 3.** (**a**) View to the eastern roof section, left: growing media Op, right: Ul-media. The line demarcating fertilizer application in June 2020 can be clearly seen, about 4 weeks after the application. (**b**) Western roof area, the fertilized area on the Op media is in the foreground while the Ulo section is in the background. The more intense blooming can be seen on 20 June 2020, four weeks after the fertilizer application.

## 3. Results

The effects of using different growing media, substrate depths, and fertilizers were significant and are demonstrated in the following section by the statistical improvements. The additional application of 10 L/m$^2$ water had no impact on the vegetation and will not be described in detail.

### 3.1. Cluster Analysis 2001–2020: Test Based on Plant Species Observation

The annual application of fertilizer affected the visual outcome as demonstrated in Figure 2.

**Test 1:** Analysis of the first year on building 3 showed the development of the plant species richness in the turf mats in Table 2. The results, supported by the cluster/correlation analyses, are shown in Table 2, with the species mix on all turf mats with no additional treatments being highly significantly similar with 0.896***, as demonstrated by the examples of the turf mats on the Ulo-West and Op-West test plots. Both mats in the year 2020 were also highly significant, with 0.963***.

**Test 2:** How did the growing media influence the number of species and the coverage value? In general, this was demonstrated here by the north test plots in 2001. The Op media in all cases showed higher species richness, as shown by way of the example in Table 3, with 25 species on the Ulo medium and 29 species on the Op medium.

At all times in 2001 and 2020, the Op media showed higher coverage values. In general, the number of observed species decreased over the past 20 years, but in contrast, the fertilizer test plots showed higher coverage values in this time.

**Table 2.** Cluster analysis: correlation based on the number of species in 2001 and 2020 on the west roof, 10-cm media. High significance between species in 2001 on both media (Ulo and Opti). Also high significance in 2020 on both media (Ulo and Opti). Also high significance in 2020 on both media. Time is more important than the different media.

| Test Plots | Ulo_West 2001 | Opti_West_ 2001 | Ulo_West 2020 | Op_West 2020 |
|---|---|---|---|---|
| Ulo_West_2001 | 1.000 | 0.896 *** | 0.151 | 0.223 |
| Ulo_West_2020 | 0.151 | 0.131 | 1.000 | 0.963 *** |

**Table 3.** Changes in the number of species from 2001 to 2020, 10-cm media.

| Name | Number of Species | Coverage Value in % |
|---|---|---|
| Ulo_Norm_N_2001 | 25 | 88 |
| Op_Norm_N_2001 | 29 | 94 |
| Ulo_Norm_Shade 2020 | 27 | 94 |
| Ulo_Fert_N_Shade_2020 | 14 | 98 |
| Ulo_Norm_N_Sun_2020 | 15 | 81 |
| Ulo_Fert_N_Sun_2020 | 13 | 96 |

In Table 4, the similarity is demonstrated by the Pearson index. Again, the similarity of the mats on the north side in 2001 was highly significant. The species similarity remained significant until 2020 in the shade on both test plots of "normal" and "fertilizer". This is demonstrated by the high grass coverage. In contrast to this, the south section differed, with higher numbers of sedum species. Shade is a significant factor determining the plant species mix.

**Test 3:** Species development of the cuttings, eastern exposure test plots.

**Table 4.** Correlation and significance (two-tailed) of the north test plots, 10-cm media, based on the number of similar plant species, in a comparison between 2001 and 2020. High significance ** on the 5‰ level between both media Ulo and Op 2001, it remain similar until 2020 on the shade plots. The sunny parts are significant different from these.

| Name | Criteria | Ulo_Norm_ N_ 2001 | Op_Norm_ N_ 2001 | Ulo_Norm_ N_Shade 2020 | Ulo_Fert_ N__Shade_ 2020 | Ulo_Norm_ N_Sun_ 2020 | Ulo_Fert_ N_Sun 2001 |
|---|---|---|---|---|---|---|---|
| Ulo_Norm_N_2001 | Pearson correlation | 1 | 0.948 ** | 0.954 ** | 0.957 ** | 0.259 | 0.210 |
| | Sig. | | 0.000 | 0.000 | 0.000 | 0.471 | 0.617 |
| | Species | 25 | 24 | 15 | 8 | 10 | 8 |

The dark material surface of Ulo has significantly higher temperatures during summer than other comparable professional roof growing media. This limited the plant growth of the seedlings and cuttings. Additionally, spontaneous plants found it difficult to establish themselves in this first year. Table 5 show the low number of species in the first year, with 9 (Ulo) and 14 (Op) species and low coverage values of 9% and 14%. These low rates were ultimately determined by the growth of *Petrorhagia saxifraga* on both east test plots, but these plants only covered a small area. In addition, the sedum cuttings must first establish themselves on the test plots. The vegetation on both areas needed a few years to achieve the estimated coverage value of 60%.

**Table 5.** Changes in the number of species on the east area with sedum cuttings from 2001 to 2020, 10-cm media.

| Name | Number of Species | Coverage Values in % |
|---|---|---|
| Ulo_Norm_East_2001 | 9 | 11 |
| Ulo_Norm_East_2020 | 16 | 69 |
| Ulo_Fert_East 2020 | 10 | 98 |
| Op_Norm_East_2001 | 14 | 14 |
| Op_Norm_East_2020 | 18 | 83 |
| Op_Fert_East_2020 | 12 | 88 |

Like the other exposure, there were differences between the fertilized and normal areas in terms of the plant species diversity. The effect of the fertilizer on all test plots is to reduce the species richness while increasing the coverage values.

Table 6 shows the correlation on the seeded eastern plots. The differences in early 2001 between the plant mix on Ulo and Op indicate widely different development. This was apparent in 2001, with very low vegetation coverage and fewer species compared to

the turf mat on the other sites. For many years, the Ulo test plots in particular had low performance values. Finally, all indicators in 2020 showed that in the long run, similar development was achieved for both media.

**Table 6.** Correlation and significance (2-tailed) of east test plots, 10-cm media, based on the number of similar plant species, in a comparison between 2001 and 2020. Significance here means: the fertilizer equalize differences between the different vegetation stands also on the east plots.

| Name | Criteria | Ulo Norm 2001 | Op Norm 2001 | Ulo Norm 2020 | Ulo Fert 2020 | Op Norm 2020 | Opt Fert 2020 |
|---|---|---|---|---|---|---|---|
| Ulo Norm 2001 | Pearson correlation | 1 | −0.359 | 1 | 1 | 1 | 1 |
| | Sig. | | 0.342 | 1 | 1 | 1 | 1 |
| | Species | 9 | 9 | 2 | 1 | 2 | 2 |
| Ulo Fert 2020 | Pearson correlation | 1 | 1 | 0.821 ** | 1 | 0.862 * | 0.869 * |
| | Sig. | | | 0.004 | 1 | 0.013 | 0.025 |
| | Species | 1 | 2 | 10 | 10 | 7 | 6 |

[1] There are not enough similar species for the correlation pairs, and significance expression cannot be calculated.

The hierarchical cluster in Figure 4 demonstrates the similarity of the eastern test plots. From the perspective of plant species development, these seeded plots are valuable observation areas. Considering the aim of achieving full coverage on the roof as soon as possible, it is important to note here that this process took nearly 10 years. These aspects must be evaluated in terms of the aims of green roofs and the risk of wind or water erosion on green roof areas that are not fully covered.

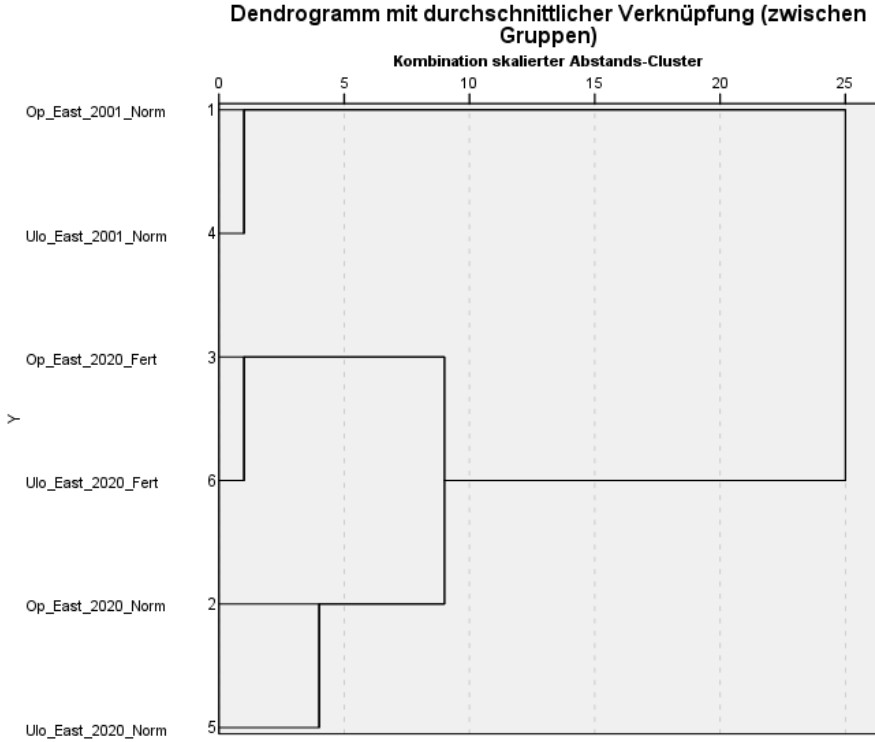

**Figure 4.** Dendrogram of the similarities between the eastern test plots from 2001 to 2020.

*3.2. Phytomass in Relation to the Type of Media and the Media Depth*

**Test 4:** focused on the phytomass on various growing media depths. The question is whether a greater growing media depth supports a higher biomass with better performance, e.g., in regard to $CO_2$ fixation.

3.2.1. The 10-cm Plots

In 2017, phytomass was harvested from all representative parts from each of the different roof orientations and exposures, with three replicates. Table 7 shows the dry matter values for the fractional biomass analysis from the vascular higher plants, mosses, and lichens. The total carbon is the sum of the shoots and roots together. All areas were fully developed over more than 10 years. One visible difference was the spontaneous growth of mosses and lichens, which represent a very special type of extensive green roof. These roofs are located in a region in northeastern Germany with clean air, and the composition of the vegetation on these roofs resembles that of typical poor sandy ground conditions, as is typical in the surrounding Müritz National Park. These dry-adapted lichens, mostly from the group of Cladonia, are endangered plants and are seldom found in ground-level habitats. The green roof is a retreat area for these plants. They are susceptible to foot traffic on dry roof conditions in summer. All values in Table 7 were calculated for an area of 1 $m^2$. In the calculation, the shoot and root phytomass of the vascular plants were calculated. The annual growth rate decreased after the establishment of full coverage with vegetation.

**Table 7.** Examples of the harvested dry phytomass in $g/m^2$ with SD from various media with 10-cm depth. Calculation for 1 $m^2$; analysis based on the mean of 3 harvesting plots each.

| Exposure | Zi [1] | Blä [1] | Opti [1] | Ulo-N-Sun [1] | Ulo-N-Shade [1] | Op-N Sun |
|---|---|---|---|---|---|---|
| Vasc.plants | 109 ± 6.5 | 183 ± 92 | 394 ± 145 | 243 ± 55 | 334 ± 88 | 411 ± 152 |
| Mosses | 1 ± 0.3 | 2 ± 0.5 | 487 ± 234 | 174 ± 135 | 846 ± 339 | 173 ± 94 |
| Lichens | 1448 ± 167 | 1080 ± 357 | 1 ± 0.3 | 444 ± 148 | 8 ± 7 | 721 ± 368 |
| Roots | 1319 ± 156 | 1456 ± 509 | 3301 ± 1740 | 2855 ± 993 | 3157 ± 305 | 2056 ± 415 |
| Dry matter [3] | 2876 ± 43 | 2719 ± 597 | 4182 ± 1656 | 3717 ± 832 | 4345 ± 314 | 3362 ± 177 |
| Total C | 1438 ± 22 | 1359 ± 298 | 2091 ± 828 | 1859 ± 416 | 2173 ± 159 | 1681 ± 88 |
| $CO_2$ | 5249 ± 79 | 4961 ± 1090 | 7633 ± 3023 | 6784 ± 1518 | 7930 ± 579 | 6135 ± 322 |

| Exposure | Op-North Shade [1] | Ulo-East [1] | Op-East |
|---|---|---|---|
| Vasc.plants | 417 ± 84 | 132 ± 48 | 461 ± 86 |
| Mosses | 534 ± 306 | 17 ± 16 | 201 ± 81 |
| Lichens | 5 ± 4 | 110 ± 37 | 229 ± 211 |
| Roots | 2940 ± 333 | 656 ± 166 | 3550 ± 965 |
| Dry matter [3] | 3896 ± 302 | 915 ± 247 | 4442 ± 1001 |
| Total C | 1948 ± 151 | 458 ± 123 | 2221 ± 500 |
| $CO_2$ | 7110 ± 550 | 1670 ± 450 | 8106 ± 1826 |

[1] Zi, Blä, Opti = on building 2, est. 1999, Op, Ulo: = on building 2, est. 2001. [3] Total dry organic matter in plants. Data from further plots are available.

Table 8 shows the correlation between vascular plant phytomass and its importance for $CO_2$ fixation. Broadly speaking, the more phytomass, the greater the effects, whereas mosses and lichens are counter-productive in that they suppress the growth of endangered higher plant species in sites with low-nutrient media while only making a minimal contribution to $CO_2$ fixation.

**Table 8.** Correlation and significance of harvested plants, n = 27 samples.

| Pearson Correlation | $CO_2$ | Vasc. Higher Plants | Mosses | Lichens | Roots |
|---|---|---|---|---|---|
| $CO_2$ fixation | 1.0 | 0.661 | 0.233 | −0.220 | 0.952 |
| Sign. $CO_2$ 1-tailed | | 0.000 | −0.121 | 0.135 | 0.000 [1] |

The dependence of $CO_2$ fixation on the phytomass follows a regression line as presented in Figure 5. More plants cover means higher $CO_2$-fixation in general.

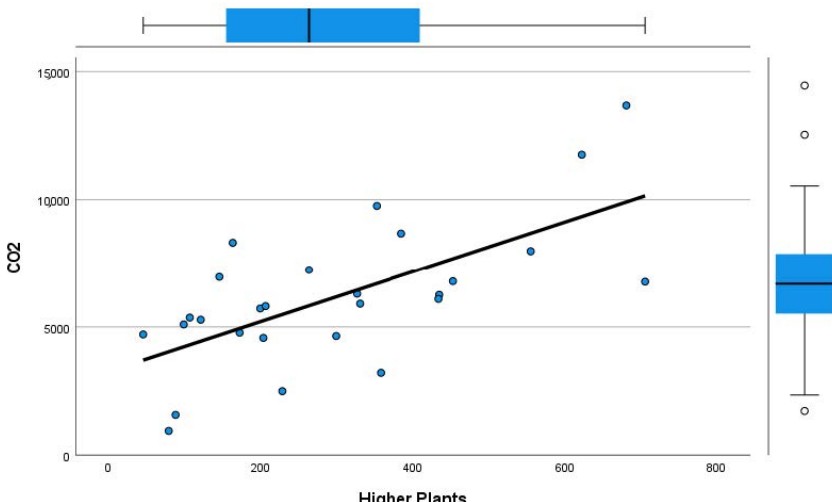

**Figure 5.** Regression—$CO_2$ fixation dependent on the phytomass of the higher plants from the 27 samples. All numbers are in grams.

The better growth rate on the 10-cm media is related to the different growing media used in these tests. Figure 6 shows the median values for $CO_2$ fixed by the plants, which varies considerably, with the lowest values of about 2000 g/m$^2$ on the Ulo media compared to average values of about 6000 g/m$^2$. In isolated cases, values that are twice as high are possible. If $CO_2$ fixation is to be the primary aim of the roof, careful choice of the media plus fertilizer is one solution. Calculating based on the approximately 8.5 million m$^2$ new green roofs realized in Germany [13], this will result in carbon storage of between 17,000 and 51,000 t $CO_2$/year after a few years.

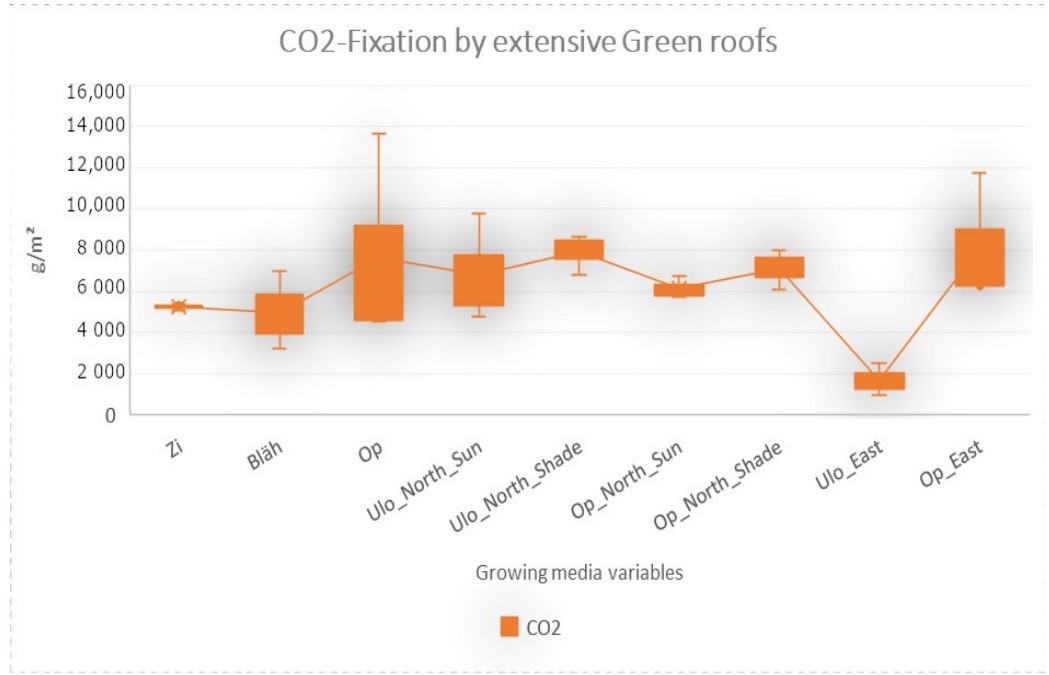

**Figure 6.** Comparison of the $CO_2$ fixation by the total phytomass (shoots and roots) of the vascular higher plants in g/m$^2$ on the nine different growing media, 10-cm layer, 3 replicates each.

### 3.2.2. The 30-cm Plots

These dry matter calculations were based on the 30-cm depth substrates, and the above-ground phytomass was harvested as described in Section 2. The 3 groups of 13 planter boxes were used in the following statistics. In the years between planting in 1999/2001 up to the first harvests in 2011 on building 2 and in 2013 on building 3, no harvesting of the plants influenced the typical growth of these plots. Working on the principle that extensive green roofs do not need maintenance, we allowed them to grow untouched for the first 10 years. The decline in the appearance was the initial reason for starting maintenance, which included mowing of the meadow. This means that the first years of harvesting, 2011 and 2013, respectively, have higher values than the annual productivity in the following years; see Table 9 with the standard deviation values.

**Table 9.** Examples of the harvested dry aboveground phytomass in $g/m^2$ and SD from the 30-cm-depth planter boxes, calculated as dry matter/$m^2$.

| Year | Media | N | x-Mean | SD |
|------|-------|---|--------|-----|
| 2011 | Op | 13 | 385 | 72 |
|      | Zi | 13 | 196 | 44 |
| 2012 | Op | 13 | 291 | 113 |
|      | Zi | 13 | 34 | 14 |
| 2013 | Op | 13 | 168 | 35 |
|      | Zi | 13 | 70 | 26 |
|      | Op-2 | 13 | 344 | 233 |
| 2014 | Op | 13 | 429 | 76 |
|      | Zi | 13 | 206 | 50 |
|      | Op-2 | 13 | 131 | 58 |
| 2015 | Op | 13 | 399 | 85 |
|      | Zi | 13 | 220 | 50 |
|      | Op-2 | 13 | 209 | 57 |
| 2017 | Op | 13 | 476 | 118 |
|      | Zi | 13 | 223 | 80 |
|      | Op-2 | 13 | 95 | 45 |

The 13 boxes with the same growing media should have similar growth rates, but as can be seen in Figure 7, the values for the phytomass vary widely. These variations are caused by several factors but are ultimately due to impacts by the users of the roof. The extensive roof growing media is poor media with no additional fertilizer. The annual aboveground productivity on this 30-cm media is between 100 and 400 $g/m^2$. The boxes on the Op media on building 2 showed the best rates. The annual productivity of the dry mass ranges between 100 and 400 $g/m^2$. The 8.5 million $m^2$ of new green roofs in Germany [13] will result in fixation of between 1551 and 6205 t $CO_2$/year.

### 3.3. Effect of Irrigation and Fertilization on Vegetation Cover and Biodiversity

In 2011, on building 3, plots were marked to create fixed areas for annual irrigation and fertilization and to compare these areas with the normal situation without any extra treatments (see Table 1).

The test questions that were to be answered with this survey were:

In the vegetation analysis, the data were interpreted as follows. Specifically, these are the coverage values for the vascular plants, the total number of species, as well as the comparison of coverage values for all sedum species. The sedum coverage is an indicator of typical extensive green roofs. The coverage of all grasses together is an indicator for the shady parts of the north side and particularly the section of the roof shaded by the elevated part of building 3.

It was observed that the performance of sedum in general decreased after the first 10 years due to a lack of fertilization and irrigation. The number of plant species

decreased on all test plots. In the following dependent paired *t*-test procedures, only the observations for 2011–2020 are interpreted.

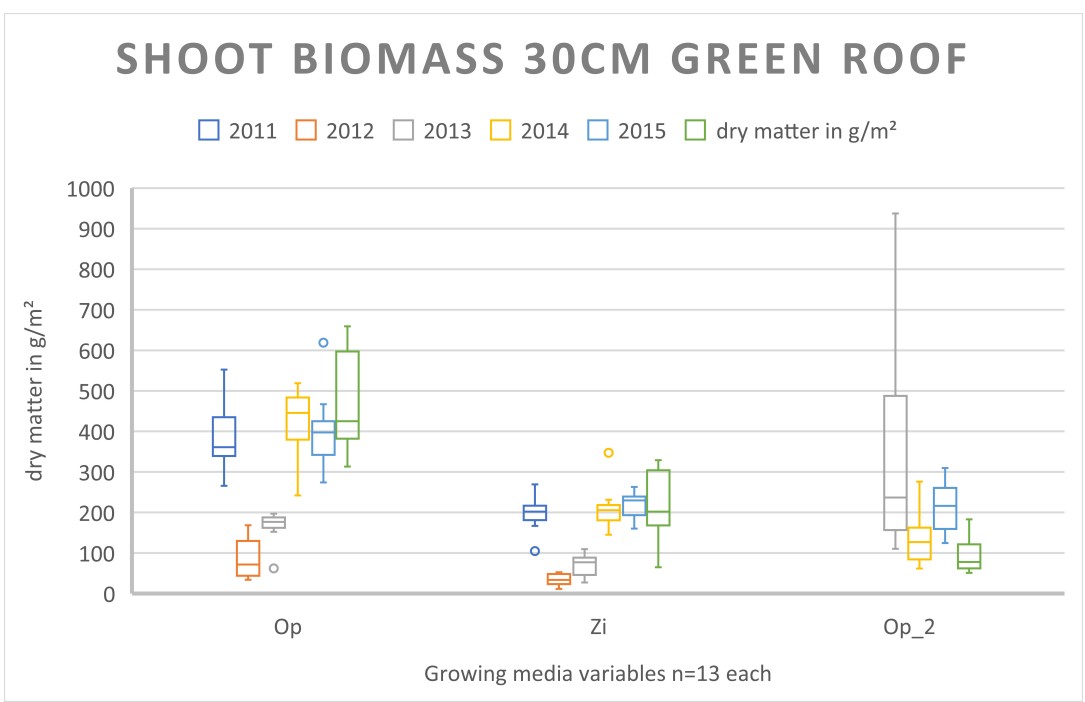

**Figure 7.** Aboveground annual harvested phytomass on the 30-cm boxes. Each of the three test plots represent 13 replicates at each date. The Op and Zi boxes are on building 2, and Op_2 is on building 3. All data refer to dry matter.

### 3.3.1. Eastern Test Plots

The eastern part of the roof with the initial plantings of sedum cuttings developed differently at all times compared to the other the test areas on the west, north, and south sides with the turf mats. In 2011, all test plots achieved the minimum value of 60%. Table 10 shows the changes in the normal areas; Ulo values were lower than the Op values. The effect of fertilizer is greater on the very poor Ulo media compared to the better performing Op media.

**Table 10.** Greening with sedum cuttings, n = 9 years, from 2011 to 2020. The coverage and number of species of the test plots with fertilizer (Fert.) and without (Norm.) for the eastern exposure. Mean value, standard deviation, and kurtosis. The negative kurtosis value indicates that the distribution is characterized by weaker marginal areas than the normal distribution.

| Criteria | East Test Plots | x-Mean | SD | Kurtosis |
|---|---|---|---|---|
| Cover | Ulo_Norm [1] | 71.6 | 11.1 | −1.9 |
| Cover | Ulo_Fert [1] | 90.2 | 6.5 | −1.1 |
| Cover | Op_Norm [1] | 90.1 | 7.9 | −1.9 |
| Cover | Op_Fert [1] | 92.3 | 6.2 | −0.2 |
| Species | Ulo_Norm [1] | 14 | 4.2 | −0.2 |
| Species | Ulo_Fert [1] | 8 | 1.5 | −1.1 |
| Species | Op_Norm [1] | 17 | 2.7 | 0.99 |
| Species | Op_Fert [1] | 10 | 2.4 | −1.2 |
| Cover Sedum | Ulo_Norm [1] | 25.5 | 8.5 | −1.2 |
| Cover Sedum | Ulo_Fert [1] | 42.6 | 5.4 | −1.1 |
| Cover Sedum | Op_Norm [1] | 32.6 | 11.9 | −1.5 |
| Cover Sedum | Op_Fert [1] | 40 | 25 | −1.1 |

[1] Ulo = Ulopor media, Op = Optima media, Fert = Fertilizer, Norm = normal without fertilizer, East = Eastern section of the research roof.

The sedum coverage values increased on both media. The number of plant species decreased with the use of fertilizer.

The following Table 11 shows the statistics for the pairwise interpretation. In general, if the T value is not 0, the significance is high. On both media (Ulo, Op), the fertilizer significantly and positively impacted the vegetation cover. In addition, the decrease in the number of species was significant. The sedum coverage was only significantly enhanced on the Ulo test plot. On Op, the sedum coverage was not significantly more extensive because of the fertilizer but they bloomed better, as can be seen in Figure 2a.

**Table 11.** Greening with sedum cuttings, n=9 years, from 2011 to 2020 (df 8), coverage and number of species for the test plots with fertilizer (Fert.) and without (Norm.) of the eastern position. Pairwise T and the level 1‰ of significance differences between fertilized and non-fertilized plots in relation to cover values on Ulo. The minus symbols in relation to the Sedum cover; a reduction of these plants in relation to other plants.

| Criteria | East Test Plots—Pairs | Mean | T | df | Significance (2-Sided) |
|---|---|---|---|---|---|
| Cover | Ulo_Fert [1] – Op_Fert [1] | −2.7 | −0.89 | 8 | 0.401 |
| Cover | Ulo_Fert [1] – Ulo_Norm [1] | 18.7 | 3.55 | 8 | 0.008 *** |
| Cover | Op_Norm [1] – Op_Fert [1] | −2.1 | −1.04 | 8 | 0.33 |
| Species | Ulo_Fert [1] – Op_Fert [1] | −1.8 | −2.1 | 8 | 0.07 |
| Species | Ulo_Norm [1] – Ulo_Fert [1] | 6 | 5.0 | 8 | 0.001 *** |
| Species | Op_Norm [1] –Op_Fert [1] | 8 | 7.7 | 8 | 0.000 *** |
| Cover Sedum | Ulo_Fert [1] – Op_Fert [1] | 2.6 | 0.32 | 8 | 0.76 |
| Cover Sedum | Ulo_Norm [1] – Ulo_Fert [1] | −17 | −4.4 | 8 | 0.002 *** |
| Cover Sedum | Op_Norm [1] – Op_Fert [1] | −7 | −1.6 | 8 | 0.146 |

[1] Ulo = Ulopor media, Op = Optima media, Fert = Fertilizer, Norm = normal without fertilizer, East = Eastern section of the research roof.

### 3.3.2. Western test plots

The turf mat layout followed the same model as the eastern test plot. The basic mean values are shown in Table 12. The differences between the fertilized and non-fertilized sections on the Ulo test plots is less dramatic, with an 11% difference compared to the eastern test plot with the cuttings with a difference of 21%. This difference is significant, as can be seen in Table 13. The reduction in the number of species is also significant for both media, as Ulo dropped from 20 to 12 and Op dropped from 27 to 16, as both tables show. Again, the sedum coverage was significantly influenced by fertilizer usage but only on the Ulo media did it result in a significant change in the coverage value.

**Table 12.** Greening with vegetation mats, n = 9 years, from 2011 to 2020. The coverage and number of species for the test plots with fertilizer (Fert.) and without (Norm.) for the western position. Mean value, standard deviation, and kurtosis.

| Criteria | West Test Plots | x-Mean | SD | Kurtosis |
|---|---|---|---|---|
| Cover | Ulo_Norm [1] | 85.1 | 6.77 | 0.09 |
| Cover | Ulo_Fert [1] | 96.4 | 4.6 | 5.7 |
| Cover | Op_Norm [1] | 92.6 | 3.6 | 0.94 |
| Cover | Op_Fert [1] | 98.1 | 0.93 | 3.3 |
| Species | Ulo_Norm [1] | 20 | 4.6 | −1.6 |
| Species | Ulo_Fert [1] | 12 | 1.7 | −1.6 |
| Species | Op_Norm [1] | 27 | 3.6 | −0.72 |
| Species | Op_Fert [1] | 16 | 2.4 | 0.72 |
| Cover Sedum | Ulo_Norm [1] | 49.9 | 2.6 | 2.6 |
| Cover Sedum | Ulo_Fert [1] | 62.4 | 17.0 | −1.3 |
| Cover Sedum | Op_Norm [1] | 50.3 | 5.6 | −2.1 |
| Cover Sedum | Op_Fert [1] | 58.8 | 10.9 | 1.9 |

[1] Ulo = Ulopor media, Op = Optima media, Fert = Fertilizer, Norm = normal without fertilizer, West = Western section of the research roof.

**Table 13.** Greening with turf mats, n = 9 years, from 2011 to 2020 (df 8), coverage and number of species for the test plots with fertilizer (Fert.) and without (Norm) of the western position. Pairwise *t*-test and the different level of significance. The significance indicate the effects of fertilizer on the cover values, negative sign, indicate reduction of the values over the years.

| Criteria | West Test Plots Pairs | Mean | T | df | Significance (2-Sided) |
|---|---|---|---|---|---|
| Cover | Ulo_Norm [1] – Op_Norm [1] | −7.4 | −2.79 | 8 | 0.024 * |
| Cover | Ulo_Fert [1] – Ulo_Norm [1] | 11.3 | 7.21 | 8 | 0.000 *** |
| Cover | Ulo_Fert [1] – Op_Fert [1] | −1.67 | −5.53 | 8 | 0.35 |
| Cover | Op_Fert [1] – Op_Norm [1] | 5.6 | 4.86 | 8 | 0.001 *** |
| Species | Ulo_Fert [1] – Op_Fert [1] | −4.2 | −4.81 | 8 | 0.001 *** |
| Species | Ulo_Fert [1-] – Ulo_Norm [1] | −8.78 | −4.25 | 8 | 0.003 *** |
| Species | Op_Fert [1] – Op_Norm [1] | −11.4 | −8.27 | 8 | 0.000 *** |
| Species | Ulo_Norm [1] – Op_Norm [1] | −6.89 | −8.04 | 8 | 0.000 *** |
| Cover Sedum | Ulo_Norm [1] –Ulo_Fert [1] | −12.56 | −2.31 | 8 | 0.050 * |
| Cover Sedum | Ulo_Fert [1] – Op_Fert [1] | 3.67 | 0.734 | 8 | 0.484 |
| Cover Sedum | Op_Norm [1] – Op_Fert [1] | −8.44 | −1.81 | 8 | 0.11 |

[1] Ulo = Ulopor, Op = Optima, Fert = Fertilizer, Norm = normal without fertilizer, West = Western section of the roof.

### 3.3.3. Northern Test Plots

In the first years of this study, the northern test plots differed between the very shady areas near the building "North_Shade" and those areas further away from the building with full sun, described here as "North Sun". This full-sun area differed from the southern test plots due to the additional heat reflected from the elevated building parts, with remarkably fewer grasses inside and a high dominance of the sedum cover. Table 14 shows that all north areas showed coverage values of above 90%. Consequently, the influence of the fertilizer was not significant in any case, as can be seen in Table 15 in the comparison of both Ulo Norm-Fertilizer pairs. Again, the reduction in the number of the plant species was significant for both growing media.

**Table 14.** Greening with vegetation mats, n = 9 years, from 2011 to 2020. Coverage and number of species for the test plots with fertilizer (Fert.) and without (Norm.) of the northern position; mean value, standard deviation, and kurtosis. Shade: in the shade of an elevated section of the roof.

| Criteria | North Test Plots | x-Mean | SD | Kurtosis |
|---|---|---|---|---|
| Cover | Ulo_Norm_Shade [1] | 91.3 | 3.24 | 0.78 |
| Cover | Ulo_Fert_Shade [1] | 91.7 | 3.67 | 0.404 |
| Cover | Op_Norm_Shade [1] | 94.3 | 3.27 | −1.68 |
| Cover | Op_Fert_Shade [1] | 98.2 | 1.19 | 0.77 |
| Species | Ulo_Norm_Shade [1] | 27 | 3.53 | 0.71 |
| Species | Ulo_Fert_Shade [1] | 16 | 3.74 | −0.11 |
| Species | Op_Norm_Shade [1] | 24 | 2.74 | 2.2 |
| Species | Op_Fert_Shade [1] | 16 | 2.6 | 0.05 |
| Cover | Ulo_Norm_Sun [1] | 91.3 | 3.24 | 0.786 |
| Cover | Ulo_Fert_Sun [1] | 90.1 | 5.01 | 0.14 |
| Cover | Op_Norm_Sun [1] | 94.6 | 2.1 | 3.0 |
| Cover | Op_Fert_Sun [1] | 97.6 | 1.24 | 1.52 |
| Species | Ulo_Norm_Sun [1] | 15 | 1.1 | 0.02 |
| Species | Ulo_Fert_Sun [1] | 11 | 1.33 | −1.97 |
| Species | Op_Norm_Sun [1] | 11 | 0.53 | −2.57 |
| Species | Op_Fert_Sun [1] | 11 | 0..53 | −2.57 |

[1] Ulo = Ulopor, Op = Optima, Fert = Fertilizer, Norm = normal without fertilizer, N = Northern section of the roof.

### 3.3.4. Southern Test Plots

The southern section of the roof is much smaller than the northern section. It gets full sun all day plus the reflection from the metal façade of the elevated section of the

building engineering area on the roof. This section of the roof also has an integrated air conditioning outlet that blows hot dry air over the southern green roof. The combination of these factors simulates extreme summer drying conditions for the green roof cover. The great performance of all the sedums growing here is remarkable. These succulents not only survive, but they also perform exceptionally well in these hot dry conditions. On the other hand, nearly all grasses are completely gone on this exposure.

The mean values of the southern plots are summarized in Table 16. With and without fertilizer, the coverage values are all above 90% and in general are the best of the test plots analyzed here. The sedum coverage values are above 70%, and with fertilizer they reach values above 80%. Again, the number of species decreased with the use of fertilizer. Table 17 presents the level of significance for these paired tests with the fertilizer test plots.

**Table 15.** Greening with turf mats, n = 9 years, from 2011 to 2020 (df 8), coverage and number of species for the test plots with fertilizer (Fert.) and without (Normal) of the northern position: North Shade: in the shade of an elevated section of the building. Sun_N: North, outside this shade. Pairwise *t*-test and the level of significance. Example explanation line 1 of Table 15: The minus sign in category "means" symbols lower cover values of the first element of the pair. As example: Athough Ulo and Op were fertilized, over the time, Ulo has all the times lower cover values than Op on the highest level of 1‰ level.

| Criteria | North Test Plot Pairs | Mean | T | df | Significance (2-Sided) |
|---|---|---|---|---|---|
| Cover | Ulo_Fert_Shade [1] –Op_Fert_Shade [1] | −6.6 | −4.8 | 8 | 0.001 *** |
| Cover | Ulo_Fert_Shade [1] – Ulo_Norm_Shade [1] | 0.33 | 1.1 | 8 | 0.347 |
| Cover | Op_Fert_Shade [1] – Op_Norm_Shade [1] | 3.89 | 3.0 | 8 | 0.017 ** |
| Cover | Ulo_Fert_Sun [1] – Op_Fert_Sun [1] | −7.4 | −4.40 | 8 | 0.002 *** |
| Cover | Ulo_Fert_Sun [1] – Ulo_Norm_Sun [1] | −1.2 | −0.53 | 8 | 0.61 |
| Cover | Op_Fert_Sun [1] – Op_Norm_Sun [1] | 3.00 | 3.84 | 8 | 0.005 *** |
| Species | Ulo_Norm_Sun [1] – Ulo_Fert_Sun [1] | 3.8 | 6.34 | 8 | 0.000 *** |
| Species | Op_Norm_Sun [1] – Op_Fert_Sun [1] | −0.11 | −0.55 | 8 | 0.594 |
| Species | Ulo_Norm_Shade [1] –Ulo_Fert_Shade [1] | 10.78 | 7.07 | 8 | 0.000 *** |
| Species | Ulo_Norm_Shade [1] – Op_Norm_Shade [1] | 3.11 | 3.18 | 8 | 0.013 ** |
| Species | Op_Norm_Shade [1] –Op_Fert_Shade [1] | 7.67 | 5.84 | 8 | 0.000 *** |
| Species | Ulo_Norm_Sun [1] – Ulo_Fert_Sun [1] | 3.78 | 6.34 | 8 | 0.000 *** |
| Species | Ulo_Norm_Sun [1] – Op_Norm_Shade [1] | 3.78 | 9.43 | 8 | 0.000 *** |
| Species | Op_Norm_Sun [1] – Op_Fert_Sun [1] | −0.11 | 0.56 | 8 | 0.594 |

[1] Ulo = Ulopor, Op = Optima, Fert = Fertilizer, Norm = normal without fertilizer, N = Northern section of the roof.

**Table 16.** Greening with vegetation mats, n = 9 years, from 2011 to 2020. Coverage and number of species, and the coverage of sedum only for the test plots with fertilizer (Fert.) and without (Norm.) for the southern position; mean value, standard deviation, and kurtosis.

| Criteria | South Test Plots | x-Mean | SD | Kurtosis |
|---|---|---|---|---|
| Cover | Ulo_Norm [1] | 92.9 | 2.4 | −2.1 |
| Cover | Ulo_Fert [1] | 98.1 | 1.69 | −1.73 |
| Cover | Op_Norm [1] | 93.7 | 2.45 | −1.14 |
| Cover | Op_Fert [1] | 96.9 | 3.2 | 1.5 |
| Species | Ulo_Norm [1] | 13 | 1.7 | −0.008 |
| Species | Ulo_Fert [1] | 9 | 2.1 | −1.91 |
| Species | Op_Norm [1] | 12 | 1.1 | −1.14 |
| Species | Op_Fert [1] | 9 | 2.12 | −1.91 |
| Cover Sedum | Ulo_Norm [1] | 74 | 5.28 | −1.08 |
| Cover Sedum | Ulo_Fert [1] | 80 | 5.5 | −1.23 |
| Cover Sedum | Op_Norm [1] | 72 | 8.12 | 1.4 |
| Cover Sedum | Op_Fert [1] | 81 | 18.4 | −0.54 |

[1] Ulo = Ulopor, Op = Optima, Fert = Fertilizer, Norm = normal without fertilizer, South = Southern section of the roof.

**Table 17.** Greening with turf mats, n=9 years, from 2011 to 2020 (df 8), coverage and number of species for the test plots with fertilizer (Fert.) and without (Norm.) of the southern position. Pairwise *t*-test and the level of significance. As explanation e.g. in line 1; also the south plots has a profit in cover value by the fertilizer, in this case on the 2‰-level. On the other hand, the fertilizer reduces the number of species. The Sedum cover is not significant influenced by the fertilizer.

| Criteria | South Test Plot Pairs | Mean | T | df | Significance (2-Sided) |
|---|---|---|---|---|---|
| Cover | Ulo_Fert [1] – Ulo_Norm [1] | 5.22 | −4.5 | 8 | 0.002 *** |
| Cover | Ulo_Fert [1] – Op_Fert [1] | 1.22 | 1.4 | 8 | 0.194 |
| Cover | Op_Norm [1] – Op_Fert [1] | −3.22 | −3.4 | 8 | 0.009 *** |
| Species | Ulo_Norm [1] – Ulo_Fert [1] | 4.0 | 7.24 | 8 | 0.000 *** |
| Species | Ulo_Norm [1] – Op_Norm [1] | 0.89 | 1.512 | 8 | 0.169 |
| Species | Op_Norm [1] – Op_Fert [1] | 1.22 | 2.82 | 8 | 0.023 * |
| Cover Sedum | Ulo_Norm [1] –Ulo_Fert [1] | 5.67 | −3.44 | 8 | 0.009 *** |
| Cover Sedum | Ulo_Fert [1] – Op_Fert [1] | −1.11 | −0.191 | 8 | 0.853 |
| Cover Sedum | Op_Norm [1] – Op_Fert [1] | −8.89 | −1.53 | 8 | 0.164 |

[1] Ulo = Ulopor, Op = Optima, Fert = Fertilizer, Norm = normal without fertilizer, South = Southern section of the roof.

## 4. Discussion

Effective solutions are needed to stop the increase in average temperatures around the globe and to reach the targets in the Paris climate agreement in the next few years [30]. On the macro-scale, as explained in Fang et al. [31], for the landmass of China, drought, temperature, and global warming are significantly connected to the existing vegetation cover. Many studies have shown that cities around the world are in general drier than their surroundings, as can be seen from examples from China [32] and 70 cities in Europe [33]. In the search for solutions on a city-wide scale, it is apparent that any decentralized greenery improves the nearby living environment of its citizens [34]. Global warming is a multi-factorial issue with an energy–water nexus. Evapotranspiration is one main energetic factor, while $CO_2$ is the accepted lead indicator from a political perspective to measure success in tackling climate change. However, not all factors can be explained by successful $CO_2$ reduction. Green roofs in general and detailed technical solutions to improve the performance of green roofs as described in this publication are methods to adapt to and mitigate climate change. On a brighter note, energetic and chemical procedures are connected and need more holistic solutions.

The economic lockdowns during the COVID-19 pandemic were reported to lead to a 7% reduction in $CO_2$ emissions in late 2020 and the start of 2021 [35]. Additional lockdowns are economically difficult and not considered acceptable for longer periods. More extensive and decentralized methods have to be considered. More $CO_2$ fixation is required, and green roofs can contribute to this. Their impact is the step from no green roof to a green roof with relatively stable values of a total amount of about 6 kg/m$^2$ as demonstrated in this survey. This value results from the total phytomass of shoots and roots of plants after a number of years of establishment, in this case 17 years. If we take this value as a baseline, extensive green roofs in Germany will result in about 51,000 t/$CO_2$ fixation per year with about 8,500,000 m$^2$ of new green roofs every year. If this annual amount needs to be increased, higher vegetation values could be achieved with a thicker layer of growing media combined with annual fertilization. This survey showed annual growth rates on 30-cm media of about 100 to 600 g/m$^2$ dry matter each year. However, this implied some maintenance of some form, such as mowing of the annual growth. Several earlier surveys measured the $CO_2$ sequestration of green roofs. Our results correspond to the wide range of sequestration values observed, which usually only focus on the aboveground material, mostly sedum [36], with variation between 64 and 381 g/m$^2$ x years. Grasses in general perform better, as can be seen in a study from Japan [21] for *Cynodon dactylon* with sequestration of 2.5 kg $CO_2$/m$^2$*year (fertilized and optimally irrigated roof modules)

and *Sedum aizoon* (non-irrigated test plot of 1.2 kg $CO_2/m^2$*year). The green roof rates are comparable to typical dry meadow habitats on the ground.

The $CO_2$ fixation is an energetic procedure to tackle global warming. Finally, the size, quality, and distribution of the green roofs are important factors to have countable effects on the city scale [37].

Climate change is resulting in longer dry periods and lower humidity. Green roofs offer a range of benefits as demonstrated by several research projects over the last 20 years [38,39]. These reviews reveal that not all the results are comparable because the methods vary widely and most of the research was conducted over short time frames.

This 20-year survey confirmed that not all results could be achieved by a single roof project. What is important is the right selection of plant species, such as that shown in Table 18, which will help to achieve full vegetation coverage with many positive ecological effects.

**Table 18.** Preferred plant mixtures on the various roof spots of this survey.

| Test Plots | North-Shade | West_East-North-Sun | South—Extreme Dry Hot Conditions |
|---|---|---|---|
| Plant preferences | Grasses | Mix of various life forms | Sedum in many variation |

Which plants are perfectly adapted to the expected climate changes? This survey was based on test plots with turf mats containing a generalized plant species mix. The test plots studied ranged from shade to extreme full sun, with some additional stress caused by high levels of solar reflection, air conditioner exhaust air outlets, and water stress.

An additional important factor is correctly choosing the right substrate. The test plot for this experiment shows that the type of growing medium is always critical for the biodiversity development over the years [40]. In most cases, highly biodiverse green roofs remain highly biodiverse over the years and support a wide range of species of birds and invertebrates [41]. As seen with this green roof research in Neubrandenburg, a significant quantity of mosses and lichens grow on very poor media. They can make up to a quarter of the total phytomass, resulting in up to 1.5 $L/m^2$ *day evapotranspiration [42]. On the one hand, the abundant presence of mosses and lichens characterizes areas with low pollution levels while also providing additional $CO_2$ fixation. On the other hand, the massive growth of mosses and lichens displaces the typical or endangered higher plants that would otherwise be expected here [43].

Semi-intensive roof construction supports a wider range of plant species, such as dry-adapted prairie plants, with higher phytomass production and ultimately higher $CO_2$ fixation [44]. The increasing number of urban agriculture sites may also be a solution for productive roof systems with massive phytomass production. Apart from the typical productive roof substrates, the competitiveness of hydroponic substrates, which have been used with increasing success in intensive green roofing, has improved in recent years [45]. The phytomass and species richness of typical extensive green roofs are comparable to dry meadows in a low mountain range [46]. Similar findings coming from North America, where more $CO_2$ could be fixed by a green roof and intensive roof gardens can be the choice [47].

This survey also shows that over a time frame of 20 years, the number of plant species significantly decreases. If this is to be avoided, some maintenance work is helpful. Additional conclusions that can be drawn from this work are:

-If a biodiverse roof is the target, variation in the growing media and media depth and additional micro niches are necessary. The research design on this roof focused on replicated research plots for statistically comparable sites.

-The study provided information about the benefits of turf mats compared to sedum cuttings.

Table 19 compares the benefits of both methods, revealing that turf mats fulfill all requirements from the beginning, but they are more expensive. The effort for the initial maintenance is quite similar. The differences are apparent in the better $CO_2$ fixation.

**Table 19.** Turf mat benefits compared to sedum cuttings as the primary vegetation cover, with results from this survey.

| Test Plots | Turf Mats | Sedum Cuttings |
|---|---|---|
| Investment | High | Low |
| Energy effort | High | Low |
| Maintenance duties | Irrigation first year | Irrigation first year |
| Vegetation coverage 60% | Immediately | After 2–3 years |
| Plant biodiversity, Year 1 | 25–29 | 9–14 |
| Plant biodiversity, Year 20 | 15–27 | 16–18 |
| $CO_2$ fixation | High | Low |

Fertilizer supports plant growth and plant performance in general [48] but not the local biodiversity. One single extra irrigation of 10 L/m$^2$ in early summer days had no significant effect on the plant performance of these dry-resistant herbs. In contrast to this, Du et al. [49] recommended emergency irrigation if shrubs are on green roofs. In times of increasing drought, the selection of the right plant species becomes an important issue. Lists with drought-tolerant plants have to be done on the regional levels, and the details in the microhabitat design support significant biodiversity [50]. Additionally, the focus on biological plant traits can help to develop an understanding regarding which species can be selected better in future times of climate change conditions [51]. In our survey, the importance of lichens of the genus *Cladonia* are highlighted as an important easy-to-care plant group and low-pollution condition, as they are in Neubrandenburg, a similar observation to a green roof working group in Canada [52]. Under the conditions of drought and climate change, prairie plants from North America become an alternative to be established in extensive green roofs under drought conditions for Central Europe [53]. Finally, the acceptance of successful growing weeds has to be accepted in natural concepts, and in many cases they can survive under extreme climatic conditions [54]. The need for local and regional biodiversity green roofs should be anchored in the related norms or guidelines in future [55].

Removing forest vegetation around the planet has significantly increased water shortages [56]. Heavier rainfall is also a consequence of the removal of vegetation in many places. This survey shows that minimal additional water has no influence on typical green roofs. Graduated tests of rain intensity and duration show the typical behavior, with extensive green roofs able to easily handle rain up to 30 L/m$^2$ [57], with the right extra drainage elements below providing support [23]. Green roofs are effective in all climate zones around the world [58]. Green roofs work like a dewfall sink in cities. In comparison to typical bitumen roofs, green roofs capture the morning dew, which helps adapted plants to survive and is also a decentralized contribution against drought in cities [59].

Green roof technology has reached the international politics against global warming. They have become elements of the renovation strategy for building stocks and the EU biodiversity strategy 2030 [60]. The results presented here can contribute to ensuring more effective green roofs as part of the future wave of renovation [61] to successfully handle the coming climate challenges. Our current knowledge of green roofs reveals opportunities to construct new quality green spaces, which can be equivalent to vegetated spaces on the ground in terms of functionality and usability [62].

As recommendations for future research, this survey has shown that there is a need for more detailed research with more variations in growing media and structures to ensure that green roof spaces are more effective instruments of green infrastructure or systems for ecosystem services [63]. Edible cities are one of the new ecosystem services to which green roof spaces can contribute to in the future [64].

In addition, it is essential that methods between research institutions are aligned so that results can be more easily compared [65]. More long-term ecosystem studies can ensure better understanding of the variation in the integrated biological solutions. As a result of global warming and the adaptation of plants to growing on artificial urban surfaces, these systems react dynamically, unlike technical solutions that remain static.

## 5. Conclusions

This study modified the typical extensive green roof with a 10-cm media depth and low maintenance in the directions of deeper growing media with 30 cm. Further treatments were fertilization and irrigation. The aim was to enhance the $CO_2$ fixation of the roof vegetation. However, what are the effects on vegetation cover and biodiversity? The summarized conclusions of these couple of years:

-Green roofs perform better if maintenance, such as extra fertilizer, are applied.

-Deeper media can capture more water, have higher vegetation cover, and finally more $CO_2$ fixation.

-On the other hand, the biodiversity decreases by higher fertilizer rates. If biodiversity is the main target of the green roof project, more habitat niches and media variation are recommended as results from this research to achieve this.

Green roofs can be established in higher quantities in all cities around the world. Although the best citizen/green roof values today exist in Stuttgart/Germany with 4 $m^2$/citizen [13], and much more is possible if more regulation in legal plans is fixed.

Green roofs are one element in the strategies against global warming and can bring back a contact to man-made nature to citizens.

Typical extensive roof vegetation is well adapted to meet future climate change challenges, but detailed planning can help to develop green roofs to meet more specific aims, such as:

-Climate-adaptive roofs with higher plant biomass.

-Biodiverse green roofs with many micro-niches for high plant species richness.

-Green roofs as elements of blue-green infrastructure with optimized water storage capacity.

-Green roofs for urban agriculture with humus media and/or hydroponic plant growth systems.

Green roofs are a small but visible factor bringing more evaporation surfaces into the anthropogenic urban desert and into the struggle against the drought.

**Author Contributions:** Conceptualization, M.K; methodology, M.K.; D.K.; Measurements M.K.; D.K.; validation, M.K.; D.K.; formal analysis, M.K.; D.K.; investigation, M.K.; D.K.; resources, M.K.; data curation, M.K.; D.K.; writing—original draft preparation, M.K.; D.K.; writing—review and editing, M.K.; D.K.; visualization, M.K.; D.K.; project administration, M.K.; All authors have read and agreed to the published version of the manuscript.

**Funding:** This research received no external funding for the research; it is part of the Green roof lab of the University of Applied Sciences. We acknowledge support for the Article Processing Charge from the Deutsche Forschungsgemeinschaft (DFG, German Research Foundation, 414051096) and the Open Access Publication Fund of the Hochschule Neubrandenburg (Neubrandenburg University of Applied Sciences).

**Informed Consent Statement:** Not applicable.

**Data Availability Statement:** The basic data sets of this research are archived in the University of Applied Sciences Neubrandenburg.

**Acknowledgments:** Special thanks to Marion Japp, Schwerin, for language improvement.

**Conflicts of Interest:** The authors declare no conflict of interest.

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
