# Peer review of "Green Roof Enhancement on Buildings of the University of Applied Sciences in Neubrandenburg (Germany) in Times of Climate Change"

_atmosphere, doi:10.3390/atmos12030382_

Round 1

Reviewer 1 Report

Title: Green roofs enhancement in times of climate change with longer dry and hot summer periods

Review: this study tested green roofs for 20 years. The paper was well-written and organized. The minor problems list below:

  1. In line 66, erase ‘.’ between species and ?.
  2. In lines 76-77, express the coordinates using degree (°), minute (‘) and second (“).
  3. In lines 24 and 87, change CO2 to CO2.
  4. In line 95, input more information of of Zi, Blä and Op including scientific names.
  5. In line 108, input references for Braun-Blanquet methods.
  6. In line 286, change ‘show’ to ‘shows’.
  7. In lines 462-463, erase ‘.’ after CO2.
  8. In lines 464-467, how large areas of the above-ground dry roof meadow means the reduction effect?
  9. In line 511, the table number is 19 not 20.

Author Response

Dear peer reviewer, 

Thank you for your comments, see here my changes, 

I the attached version 1, all further changes are visible by the track changes.

The attached document, with track changes show all further changes.

Title: Green roofs enhancement in times of climate change with longer dry and hot summer periods

Review: this study tested green roofs for 20 years. The paper was well-written and organized. The minor problems list below:

  1. In line 66, erase ‘.’ between species and ?.    done
  2. In lines 76-77, express the coordinates using degree (°), minute (‘) and second (“).   

The details of the roofs can be seen on Google maps with the GPS Coordinates: Degree:Minutes:Seconds); see the sites: building 2: 53:33:23N 13:14:44E and building 3: 53: 33:15N 13:14:43E

  1. In lines 24 and 87, change CO2 to CO2.  Done
  2. In line 95, input more information of of Zi, Blä and Op including scientific names.

Building 2: With three commercial growing media, identified as Zi, Blä, and Op, with 10 cm depth plus a drainage layer. The additional 26 planter boxes with 30-cm thick layer of the extensive growing media from two main deliverer of Green roof materials ZinCo and Optigruen, with the abbreviation Zi and Op are the test installations. These materials follow the all the requirements after [16], such as granulometric distribution and the minimum water holding capacity. The primary vegetation was similar using grass seeds and sedum cuttings.

-Building 3: The 10-cm growing media Op-2 (product name “Optima Tiefgarage schwer”) and Ulo (expanded slate, grain size 2–11 mm, brand name “Thüringer Blähschiefer”).

  1. In line 108, input references for Braun-Blanquet methods. Done

Braun-Blanquet methods [25]

Kopecký, K.; Dostálek, J.; Frantík, T. The use of the deductive method of syntaxonomic classification in the system of vegetational units of the Braun-Blanquet approach. Vegetatio 1995, 117, p. 95–112. https://doi.org/10.1007/BF00045502

Best,

Manfred Koehler 

  1. In line 286, change ‘show’ to ‘shows’.  done
  2. In lines 462-463, erase ‘.’ after CO2.    done
  3. In lines 464-467, how large areas of the above-ground dry roof meadow means the reduction effect?   Related to one squaremeter

  1. In line 511, the table number is 19 not 20. Done

Author Response

The structure of the article is correct, clearly developed and with an adequate scientific level.

The methodology used is appropriate, with a detailed description of the procedure.

The results are presented and explained with precision accompanied by numerous tables and figures.

In general, it is of sufficient quality for publication.

However, we offer the following suggestions: -The Conclusions must expressly include a synthesis of the set of results obtained in this research.

We expanded the Introduction, the conclusion and the discussion. Also added about 20 relevant related publication to compare our results….

It is convenient to offer a simplified summary of the various information already provided and highlight your contribution.

Some conclusion had be simplified, basically the information: Of course, forest is the most important theme to struggle against climate change, but our methods are connected to the environment where the majority of people live (in cities), AND every squaremeter of greener counts ….

-It is a local investigation, so the authors should indicate if, in their opinion, the experience can be extrapolated to other climatic areas.

Local survey, yes added the name of our research institution into the title .

 - Personally, we consider that the contribution of green roofs to mitigating climate change is positive but extremely limited.

This topic is covered by expansion of the introduction see:

. Introduction

Drought is related to massive deforestation in many parts of the world [1]. The daily new ground sealing around the world are not stopped, but new tree plantation initiatives work against this trend to stop the further dry out of our planet. Successful initiative of plantings are active from Australia, under the term of “Land-care” [2], in Mongolia [3] or the current Green Wall movement [4] in North Africa. It will be a long way to reach forest cover values, such as existed e.g. in Central Europe 2000years ago with above 80% in contrast to today with about 30% forest cover [5]. The increasing world population is one reason for the deforestation [6]. Drought in Cities is connected with the low amount of evaporative green areas in cities [7, 8]. The consequences are longer hot and dry summer periods that cause a number of environmental and health problems for residents. According to Kravcik et al. [9], evaporative surfaces are key instruments to combat global warming in cities. Wherever it is possible, urban forestry should be a first choice to enhance the ecosystem services by planting trees, like the “trillion tree initiative” [10]. However, a second best choice is to green building surfaces, which are normally un-vegetated.

Compare the agricultural and forest area of a region with the possible use of green roofs in buildings.

Agriculture solutions are important but also inner city activities with a lack in greenery in nearly all Cities in the world.

The discussion section presented by the authors is excessively general, so it is necessary to clarify these aspects.

We added relevant further related publication, and come to the conclusion of current policy activities in the EU.

All track changes are visible in the added version .

Thank you for your valuable comment.

Manfred Koehler

Reviewer 3 Report

The present study is an interesting research about Green roofs, Phytomass, biodiversity and urban climate on two buildings of the University of Applied Sciences in Neubrandenburg (Germany).

Some considerations about the article:

1.- The title does not reflect the content of the article. It is too general and generates too global and climatic expectations. For this reason, in my opinion, the title should collect more specifically the analyzed environment and the subject work. I consider that this is not a simple formal aspect.

2.- The writing of the text is sometimes more like a technical report than a research article. In the section "Research design" this question is clearly manifested.

3.- The introduction section is quite poor. The authors should delve further into the key aspects of their work: evaporative phytomass produced on green roofs, irrigation or fertilization and plant biodiversity on roofs, and green roofs and CO2 fixation.

4.- In the discussion section the authors should consider results obtained in other works

Author Response

The present study is an interesting research about Green roofs, Phytomass, biodiversity and urban climate on two buildings of the University of Applied Sciences in Neubrandenburg (Germany).

Some considerations about the article:

1.- The title does not reflect the content of the article. It is too general and generates too global and climatic expectations. For this reason, in my opinion, the title should collect more specifically the analyzed environment and the subject work. I consider that this is not a simple formal aspect.

We added to the title the direct location.

2.- The writing of the text is sometimes more like a technical report than a research article. In the section "Research design" this question is clearly manifested.

The chapter Research design is updated with one more chapter to make the scientific side more visible:  

Research design

In contrast to many other green roof research studies, the focus here is the long time performance of typical FLL-Standard [24] green roofs under real roof conditions. The second aspect is a complete green roof not only small test installation. Easy accessibility and the inclusion into teaching programs open a frequent observation of changes. The use of market leader products, as in the German FLL-guidelines since the 1990th strongly recommended helps to improve these technical standards. Additionally some further growing media and treatments, such as irrigation, fertilization helps to move forward the existing knowledge. At the beginning in the year 1999 basic aims was, to achieve a 60% vegetation cover on very shallow layer. In the last years, new upcoming questions were the enhancement of biodiversity and the CO2-fixation under hotter and dryer summer periods in Central Europe. The basic research design can be described on both buildings as follows:

3.- The introduction section is quite poor. The authors should delve further into the key aspects of their work: evaporative phytomass produced on green roofs, irrigation or fertilization and plant biodiversity on roofs, and green roofs and CO2 fixation.

The introduction is now much more specific, about 50% longer and more relation to the research question. See the track change version attached.

4.- In the discussion section the authors should consider results obtained in other works

The discussion section is also nearly doubled and more specific.

In general: the references is updated with 20 more related works.

Thank you for your valuable comments, that helped

Best,

Manfred Koehler

Round 2

Reviewer 3 Report

In general, the authors have attended the requirements that I presented in the first round of review. Therefore, I believe that the article can be published in the magazine without further consideration on my part.